

# Sharing diverse information gets driver agents to learn faster: an application in en route trip building

Guilherme Dytz dos Santos and Ana L.C. Bazzan

Computer Science, UFRGS (Universidade Federal do Rio Grande do Sul), Porto Alegre, RS, Brazil

## ABSTRACT

With the increase in the use of private transportation, developing more efficient ways to distribute routes in a traffic network has become more and more important. Several attempts to address this issue have already been proposed, either by using a central authority to assign routes to the vehicles, or by means of a learning process where drivers select their best routes based on their previous experiences. The present work addresses a way to connect reinforcement learning to new technologies such as car-to-infrastructure communication in order to augment the drivers knowledge in an attempt to accelerate the learning process. Our method was compared to both a classical, iterative approach, as well as to standard reinforcement learning without communication. Results show that our method outperforms both of them. Further, we have performed robustness tests, by allowing messages to be lost, and by reducing the storage capacity of the communication devices. We were able to show that our method is not only tolerant to information loss, but also points out to improved performance when not all agents get the same information. Hence, we stress the fact that, before deploying communication in urban scenarios, it is necessary to take into consideration that the quality and diversity of information shared are key aspects.

## INTRODUCTION

With the COVID-19 related pandemic, there has been several reports that the use of private transportation means (e.g., individual vehicles) is increasing as people try to avoid public transit as much as possible. This leads to even more congestion and hence makes the question of selecting a route to go from A to B more and more prominent. This is especially the case for commuters, who make a given trip nearly every day and, hence, have the opportunity to learn and/or adapt to the traffic patterns faced daily. To address the challenges posed by an ever increasing demand, transportation authorities and traffic experts try to distribute the flow among existing routes in order to minimize the overall travel time. Often, this task involves some form of communication with the drivers. Traditional approaches such as variable message panels or radio broadcast are now being replaced by directed (and potentially personalized) communication, via new kinds of communication devices.

Corresponding author
Ana L.C. Bazzan,
bazzan@inf.ufrgs.br

While the current pattern is that each individual driver selects a route based on his/her own experience, this is changing as new technologies allow all sorts of information exchange. Examples of these technologies are not only based on broadcast (e.g., GPS or cellphone information) but also a two-way communication channel, where drivers not only receive traffic information but also provide them. Hence, currently, many traffic-related applications for cellphones deal with the idea of a central authority in charge of somehow assigning routes for drivers. Examples are Waze, Google apps, etc. Since their specific algorithms are not published, one can only guess that they try to find a feasible solution, given a set of constraints that they are able to infer from the current data they collect. What seems certain is that these platforms work in a centralized way, based on data they collect when their customers or users use their specific apps. Also, they do not handle locally collected and processed data. This leads to them being ineffective when the penetration of their services is low as, for example, during the initial stages of the 2020 pandemics, when few drivers were using the system. A way to mitigate this could be to decentralize the processing of information, as proposed here, and passing it to drivers to make their route choices.

Our method has some resemblance with the notion of traffic assignment (see "Background and Related Work"), since it is based on the fact that drivers collect experience by trying out several routes until they settle on those that lead to the least travel time.

Traffic assignment approaches work (and indeed were developed for this purpose) well for *planning tasks*, that is, how to plan a traffic network (or change an existing one) in order to minimize travel costs. However, route choice is not related to planning tasks but, rather, is an operational aspect, especially in commuting situations, where drivers repeatedly travel from the same origin to the same destination. Besides, traffic assignment is a centralized approach, in which the drivers do not actively select routes. Rather, routes are assigned to them. Thus, it is important to investigate how drivers do select routes in their daily commuting tasks.

Multi-agent reinforcement learning (MARL) can be used for such purpose, as it fits the task of letting agents decide, autonomously, how to select routes to go from A to B. This is realized by letting agents iteratively choose their least costly route based on their own learning experiences. Such approach has been tried before, as described in the section on related works. In fact, it has been shown that reinforcement learning is a good technique to investigate route choice. However, the learning process can be inefficient, as for instance, it may take time, since the agents have to collect experiences by themselves. As this happens to be a very noisy environment, the signal an agent gets can be little discriminatory (e.g., due to the presence of other learning agents, an agent may get the same signal for very different actions, or, conversely, different signals for the same action). Thus, our long term aim is to investigate forms of accelerating the learning process. One of these forms is by giving more information to the agents. There are only few works that consider new technologies to this experience, as for instance those tied to vehicular communication in general.

In the present article, we extend a method that connects MARL to new technologies such as car-to-infrastructure communication (C2I). These were formulated with the goal

of investigating how C2I communication could act to augment the information drivers use in their learning processes associated with choices of routes. In such approach, whole routes are not imposed or recommended to drivers, but rather, these receive local information about the most updated state of the links that happen to be near their current location. This way, drivers can change their route on-the-fly (the so-called en route trip building). Further, that approach assumes that the infrastructure is able to communicate with the vehicles, both collecting information about their most recent travel times (on given links), as well as providing them with information that was collected from other vehicles. However, another assumption is that messages are never lost, which is not realistic. Thus, in the present article, we relax this assumption and admit loses of messages, as well as investigate the impact of them on the overall performance.

As a result of such extension, we are able to confirm that the MARL technique combined with a C2I model can accelerate the learning process. Moreover, our approach is tolerant to information loses.

In short, the contribution of the present work is manifold. First, we employ MARL to the task of learning how to go from A to B. Second, we do this using a non trivial scenario (as it is the case in most of the literature), in which there are more than one origin-destination pair. Third, we depart from most of the literature where the learning task considers that the driver agents already know a set of (pre-computed) routes to select among. Rather, we let these agents build their trips en route. This in turn requires the use of a microscopic, agent-based approach, where agents can potentially use different pieces of information in order to perform en route choice. This again contrasts to most of the literature, which uses macroscopic modeling (e.g., by means of abstract cost functions to compute travel times). Fourth, we connect MARL with the aforementioned communication technologies, in order to investigate whether the learning process can be accelerated by exchange of local information only. Lastly, we extend a previous approach by investigating its robustness to loses of messages.

This article is organized as follows. The "Background and Related Work" briefly presents some background concepts on traffic assignment and reinforcement learning, as well as the panorama on the related work. Following, our methods and experimental results are presented and discussed. We review the general conclusions and outline the future work in the last section.

# BACKGROUND AND RELATED WORK

## The traffic assignment problem

In transportation, the traffic assignment problem (TAP) refers to how to connect a supply (traffic infrastructure) to its demand, so that the travel time of vehicles driving within a network is reduced. This network can be seen as a graph $G = (N, E)$, where $N$ is the set of nodes that operate as junctions/intersections, and $E$ is a set of directed links (or edges, as both terms are used interchangeably) that connect the nodes. Hence the goal is then to assign vehicles to routes so that the travel time is minimized.

For more details, the reader is referred to Chapter 10 in *Ortúzar & Willumsen (2011)*. For our purposes it suffices to mention that classical approaches aim at planning tasks, are

centralized (i.e., trips are *assigned* by a central authority, not *selected* by individual drivers). Also, the main approaches are based on iterative methods that seeks convergence to the user equilibrium (see "User Equilibrium").

## User equilibrium

When it comes to reaching a solution to the TAP, one can take into account two perspectives: one that considers the system as a whole, and one that considers each user's point of view. In the system perspective, the best solution refers to the system reaching the best average travel time possible; this is the so called system optimum (SO), or Wardrop's second principle (*Wardrop, 1952*). We stress that the SO is a desirable property, but hardly achievable given that it comes at the cost of some users, who are not able to select a route leading to their personal best travel times.

On the other hand, and most relevant for our current work, at the user's perspective, the system reaches the user (or Nash) equilibrium (UE) when there is no advantage for any individual to change its routes in order to minimize their travel time, as stated in the first Wardrop's principle (*Wardrop, 1952*). The UE can be achieved by means of reinforcement learning, as discussed next.

## Reinforcement learning

Reinforcement learning (RL) is a machine learning method whose main objective is to make agents learn a policy, that is, how to map a given state to a given action, by means of a value function. RL can be modeled as a Markov decision process (MDP), where there is a set of states $S$, a set of actions $A$, a reward function $R : S \times A \rightarrow \mathbb{R}$, and a probabilistic state transition function $T(s, a, s') \rightarrow [0, 1]$, where $s \in S$ is a state the agent is currently in, $a \in A$ is the action the agent takes, and $s' \in S$ is a state the agent might end up, taking action $a$ in state $s$, so the tuple $(s, a, s', r)$ states that an agent was in state $s$, then took action $a$, ended up in state $s'$ and received a reward $r$. The key idea of RL is to find an optimal policy $\pi^*$, which maps states to actions in a way that maximizes future reward.

Reinforcement learning methods fall within two main categories: model-based and model-free. While in the model-based approaches the reward function and the state transition are known, in the model-free case, the agents learn $R$ and $T$ by interacting with an environment. One method that is frequently used in many applications is Q-Learning (*Watkins & Dayan, 1992*), which is a model-free approach.

In Q-learning, the agent keeps a table of $Q$-values that estimate how good it is for it to take an action $a$ in state $s$, in other words, a $Q$-value $Q(s, a)$ holds the maximum discounted value of going from state $s$, taking an action $a$ and keep going through an optimal policy. In each learning episode, the agents update their $Q$-values using the Eq. (1), where $\alpha$ and $\gamma$ are, respectively, the learning rate and the discounting factor for future values.

$$Q(s, a) = Q(s, a) + \alpha(r + \gamma max_a[Q(s', a') - Q(s, a)]) \tag{1}$$

In a RL task, it is also important to define how the agent selects actions, while also exploring the environment. A common action selection strategy is the ε-greedy, in which

the agent chooses to follow the optimal values with a probability $1 - \varepsilon$, and takes a random action with a probability $\varepsilon$.

While this basic approach also works in MARL, it is important to stress some challenging issues that arise in an environment where multiple agents are learning simultaneously. Complicating issues arise firstly due to the fact that while one agent is trying to model the environment (other agents included), the others are doing the same and potentially changing the environment they share. Hence the environment is inherently non-stationary. In this case, convergence guarantees, as previously known from single agent reinforcement learning (e.g., *Watkins & Dayan (1992)* regarding Q-learning), no longer hold.

A further issue in multi-agent reinforcement learning is the fact that aligning the optimum of the system (from the perspective of a central authority) and the optimum of each agent in a multi-agent system is even more complicated when there is a high number of agents interacting.

## Related work

Solving the TAP is not a new problem; there have been several works that aim at solving it. In one front, there are have classical methods (see Chapter 10 in *Ortúzar & Willumsen (2011)*), which, as aforementioned, mostly deal with planning tasks. Further, the TAP can also be solved by imposing tolls on drivers (*Sharon et al., 2017*; *Buriol et al., 2010*; *Tavares & Bazzan, 2014*). The latter specifically connects road pricing with RL. However, the focus is on learning which prices to charge. Besides these two fronts, RL for route choice is turning popular.

When we refer to RL methods to solve the TAP, these usually fall into two categories: a traditional RL method, and a stateless one. Contrarily to the traditional approach, in the stateless case, the agents actually have only one state that is associated with its origin-destination pair, and they choose which actions to take. Actions here correspond to the selection of one among $k$ pre-computed routes. Works in this category are *Ramos & Grunitzki (2015)* (using a learning automata approach), and *Grunitzki & Bazzan (2017)* (using Q-learning). In *Zhou et al. (2020)* the authors used a learning automata approach combined with a congestion game to reach the UE. *Tumer, Welch & Agogino (2008)* adds a reward shaping component (difference utilities) to Q-learning, aiming at aligning the UE to a socially efficient solution.

Apart from the stateless formulation, in the traditional case, agents may found themselves in multiple states, which are normally the nodes (intersections) of the network. Actions then correspond to the selection of one particular link (edge) that leaves that node. In *Bazzan & Grunitzki (2016)* this is used to allow agents to learn how to build routes. However, they use a macroscopic perspective by means of cost functions that compute the abstract travel time. In the present article, the actual travel time is computed by means of a microscopic simulator (details ahead). A microscopic approach is required to handle communication issues.

As aforementioned, our approach also includes C2I communication, as these kinds of new technologies may lead agents to benefit from sharing their experiences (in terms of

travel times), thus reducing the time needed to explore, as stated in *Tan (1993)*. The use of communication in transportation systems, as proposed in the present paper, has also been studied previously (*Grunitzki & Bazzan, 2016*; *Bazzan, Fehler & Klügl, 2006*; *Koster et al., 2013*; *Auld, Verbas & Stinson, 2019*). However, these works handle communication at abstract levels, using macroscopic approaches. In some cases, the information is manipulated to bias the agents to reach an expected outcome. Moreover, most of these works deal with vehicular communication (i.e., messages are shared among the vehicles), or are based on broadcast of messages by one or few entities. This scheme approaches either systems such as traffic apps we see nowadays (Waze, etc.), or messages distributed by the traffic authority (as it used to be the case some time ago, using radio or variable message panels on main roads as in *Wahle et al. (2000)*). Neither vehicular communication nor broadcast are appropriate to investigate the impact of sharing local information, as we do here. A previous work by us (*Santos & Bazzan, 2020*) has presented preliminary results about the performance of combining RL with C2I against RL without communication. However, in this work, it is assumed that messages exchanged among the various actors do not get lost, which is irrealistic. Therefore, in the present article we focus on the impact of communication failure and also on what type of information yields better results.

In a different perspective, works such as *Yu, Han & Ochieng (2020)* evaluate the impact of incomplete information sharing in the TAP. They do not employ a RL-based but rather a classical approach, namely multinomial Logit model.

More recently, *Bazzan & Klügl (2020)* discuss the effects of a travel app, in which driver agents share their experiences. The idea is to "mimic" what happens in an office where colleagues chat about their habits and route choice experiences. In the present article, driver agents do not directly share their experiences since the work in *Bazzan & Klügl (2020)* has shown that this process may lead to sub-optimal results, due to agents not taking local issues into account. This is hardly possible in that work since *Bazzan & Klügl (2020)* use a macroscopic simulator, where location is an abstract concept. Rather, the present paper proposes—as shown in the "Methods"—that the information is exchanged via an intersection manager, that is, a manager of a portion of the network.

In any case, this sharing of knowledge was proposed in other scenarios (*Tan, 1993*) and refers generally to the research on transfer learning (*Taylor et al., 2014*; *Torrey & Taylor, 2013*; *Fachantidis, Taylor & Vlahavas, 2019*; *Zimmer, Viappiani & Weng, 2014*). It is important to note though, that virtually all these works deal with cooperative environments, where it makes sense to transfer knowledge. In non-cooperative learning tasks, as it is the case of route choice, naive transfer of learned policies may lead to every agent behaving the same, which runs against the notion of efficient distribution of agents in the road network.

## METHODS

Our approach is based on using communication to augment the information each agent[1] has and, hence, the learning performance. The next three subsections discuss, respectively: how the infrastructure is represented; how communication occurs; and the details of the RL algorithm. We then formalize the details as an algorithm.

[1] Henceforth, the term agent is used to refer to a vehicle and/or driver agent.

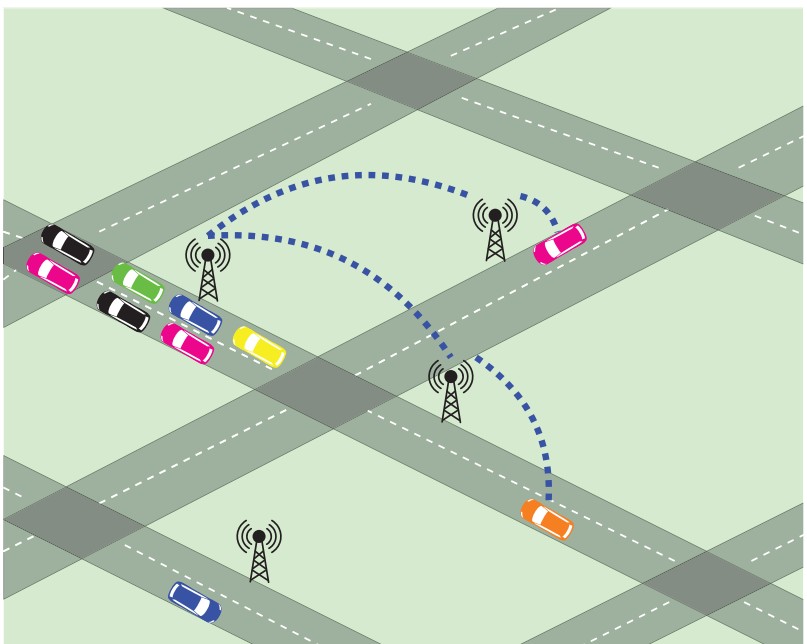

**Figure 1 Scheme of the communication infrastructure.** This figure was designed using assets from https://www.vectorportal.com/ and https://www.freepik.com. All assets used fall under license CC BY 4.0.

## Representing the infrastructure

We assume that every node $n \in N$ present in the network $G$ is equipped with a communication device (henceforth, CommDev) that is able to send and receive messages in a short range signal (e.g., with vehicles around the intersection). Figure 1 shows an scheme that represents $G$ and CommDevs within $G$.

Using the short-range signal, the CommDevs are able to communicate with vehicles that are close enough, and are able to exchange information related to local traffic data (refer to next section for details). Moreover, these CommDevs are able to store the data exchanged with the agents in order to propagate this information to other agents that may use nearby intersections in the near future.

The arrows that connect CommDevs in Fig. 1 represent a planar graph, meaning that every CommDev is connected and can communicate to its neighboring devices. This permits that CommDevs get information about the traffic situation in neighboring edges, which is then passed to the agents.

## How communication works

Every time an agent reaches an intersection, prior to choosing an action (the next intersection to visit), it communicates with the intersection's CommDev (see Fig. 1) to exchange information. The actual piece of information sent from agents to CommDevs is travel times (hence, rewards) received by the agents, regarding their last action performed.

Conversely, the infrastructure communicates to the agent information about the state of the nearby edges, in terms of which rewards an agent can expect if it selects to use that particular link. This information can be of various forms. In all cases, the expected

reward is computed by taking into account the rewards informed by other agents, when they have used nearby links. In the experiments, we show results where CommDevs communicate expected rewards that are either an aggregation (over a time window) or just a single value.

In any of these cases, an agent receiving such information will then take it into account when selecting an action (choice of a link) in that particular state (a node). Next, details about how the information is processed, by both the CommDevs and the vehicle agents, are given.

### Information hold by infrastructure

Each CommDev uses queue based data structures to hold the rewards informed by each agent that passes through it. Specifically, each edge is associated with one data queue. These queues have a maximum size, and when new information arrives after the queue is full, the oldest reward stored is discarded to make room to the most recent one.

When an agent requests information, the CommDev retrieves the rewards collected for the agent's possible actions and passes it to that agent. Recall that an action corresponds to a link to be traveled next, in order to form a route to the agent's destination.

### Information used by the agent

In a standard Q-learning algorithm, the agents update their Q-values based on the feedback from the action they have just taken. However, in our case agents also update their Q-values based on the expected rewards received by the infrastructure. This means that every time they reach an intersection, they update their Q-values with the information provided by the CommDevs. We do this in order to accelerate the learning process. Instead of just considering its own past experiences, the information provided by the CommDevs augment the knowledge each agent has.

It is worth noting that a distinguishing characteristic of our approach is that it deals with local information, thus the information received from the CommDev only concerns actions that can be selected from that particular node.

Given a network $G$, every agent (vehicle) $v \in V$ has a pair $(o, d) \in N \times N$, that defines its origin-destination pair (OD-pair). Nodes $n \in N$ are seen as states the agents might be in, and the outgoing edges of a node $n$ are the possible actions for that given state. Hence, the agents build their routes on-the-fly by visiting nodes and edges.

Upon choosing an action (edge) $e$, $v$ perceives its reward. We recall that being a microscopic model, this reward is actually computed by the simulator, rather than by an abstract cost function, as it would be the case in a macroscopic model.

Assuming that the simulator reports a travel time of $t_e^v$ for agent $v$ traveling on edge $e$, the reward is $-t_e^v$, as we want to make sure the agents prefer to take edges that minimize travel times.

This alone does not guarantee that the agents will reach their destination fast, as they might end up running in loops throughout the network. Hence a positive bonus $B$ is given to each agent that reaches its destination, giving them incentives to end their trips as fast as possible.

---

**Algorithm 1** Q-learning with C2I.

1: **Input:** *G, D, P, M, α, γ, ε, B*

2: *s←0*

3: **while** *s < M* **do**

4:    **for** *v* in *V* **do**

5:     **if** *v.finished_trip()* **then**

6:       *v.update_Q_table(B−v.last_edge_travel_time)*

7:       *G.commDev[v.curr_node].update_queue(v.last_reward, v.last_edge)*

8:       *v.start_new_commuting_trip()*

9:     **else if** *v.has_reached_a_node()* **then**

10:       *v.update_Q_table(−v.last_edge_travel_time)*

11:       *G.commDev[v.curr_node].update_queue(v.last_reward, v.last_edge)*

12:       *v.update_Q_values(G.commDev[v.curr_node].info)*

13:       *v.choose_action()*

14:    **end if**

15:    **end for**

16:    *s←s+1*

17: **end while**

---

We deal with a commuting scenario, where each agent performs day-to-day experiments in order to reach an equilibrium situation, in which no agent can reduce its travel time by changing routes. Because agents belong to different OD pairs and/or select different routes, their trips take different number of simulation steps. These steps represent elapsed seconds in simulation time. Hence, this means that not every agent finishes its trip simultaneously and, therefore, the standard notion of a learning episode cannot be used here. Rather, each agent has its own learning episode that will take as many simulation steps as necessary to reach its destination.

Next, we explain the main parts of our approach, which can be seen in Algorithm 1.

Line 1 list the inputs of Algorithm 1: *G* is the topology of the network, *D* is the demand (flow rate) that is inserted in the network, *P* is the set of OD-pairs, and *M* is the maximum number of steps to simulate. It is also necessary to set *α*, *γ* (both relating to Eq. (1)), *ε* for controlling the action selection and the exploration-exploitation strategy, and the bonus *B*.

The main loop is presented between lines 3 and 17, where the learning and the communication actually take place. The first *if* statement shown in line 5 takes care of all agents that finished their trips in the current step: agents perceive their reward plus the bonus for finishing the trip. At Line 7, each agent informs the corresponding CommDevs the rewards, and since its trip has ended, it gets reinserted at the origin node to start a new learning episode (as this is a commuting scenario).

The *if* statement at line 9 represents the intermediary nodes, where each agent also perceives its reward and informs the CommDev (line 11) about the reward just

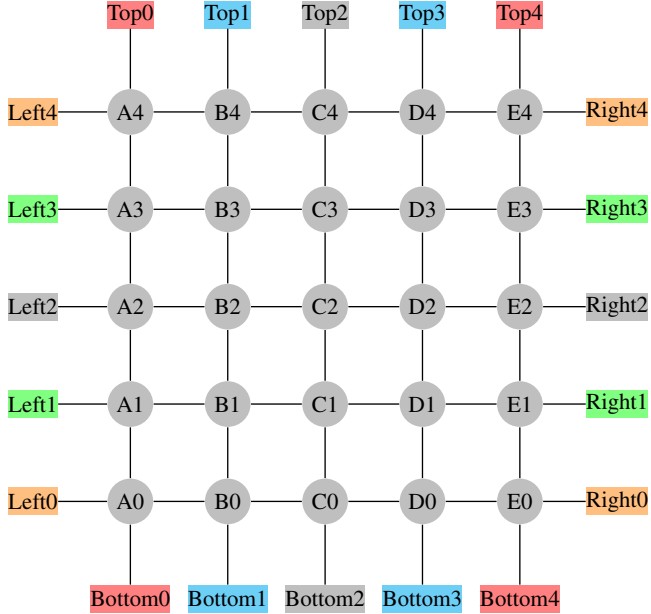

**Figure 2 5 × 5 Grid Network.**

experienced, so that the CommDev can update its queue structure. In line 7, each agent updates its *Q*-value for the last action based on its own experience, that is, with the actual reward received for traveling through the last link.

Following, a CommDev also informs agents about the rewards that can be expected from the actions each agent might take next (line 12). Each agent then updates its Q-table and chooses an action.

# EXPERIMENTS, RESULTS, AND ANALYSIS

## Scenario: network and demand

Simulations were performed using a microscopic tool called Simulation of Urban Mobility (SUMO, *Lopez et al. (2018)*). SUMO's API was used to allow vehicle agents to interact with the simulator en route, that is, during simulation time.

The scenario chosen is a 5 × 5 grid depicted in Fig. 2; each line in the figure represents bi-directed edges containing two lanes, one for each traffic direction. It is also worth noting that each directed edge is 200 m long.

The demand was set to maintain the network populated at around 20–30% of its maximum capacity, which is considered a medium to high density. Recall that no real-world network is fully occupied at all times, and that the just mentioned density level does not mean that there will not be edges fully occupied, which happens from time to time; this percentage is just the average over all 50 edges.

This demand was then distributed between the OD-pairs as represented in Table 1. The last column represents the volume of vehicles per OD-pair. Those values were selected so that the shorter the path, the smaller the demand, which seems to be a more realistic assumption than a uniform distribution of the demand.

**Table 1 Demand per OD-pair.**

| Origin | Destination | Demand |
|---|---|---|
| Bottom0 | Top4 | 102 |
| Bottom1 | Top3 | 86 |
| Bottom3 | Top1 | 86 |
| Bottom4 | Top0 | 102 |
| Left0 | Right4 | 102 |
| Left1 | Right3 | 86 |
| Left3 | Right1 | 86 |
| Left4 | Right0 | 102 |

Two points are worth reinforcing here. First, vehicles get reinserted at their corresponding origin nodes, so that we are able to keep a roughly constant insertion rate of vehicles in the network, per OD pair. However, this does not mean that the flow *per link* is constant, since the choice of which link to take varies a lot from vehicle to vehicle, and from time to time. Second, despite being a synthetic grid network, it is not trivial, since it has 8 OD pairs, which makes the problem complex as routes from each OD pair are coupled with others. As seen in Table 1, we have also increased such coupling by designing the OD pairs so that all routes traverse the network, thus increasing the demand for using the central links.

### Q-learning parameters

A study conducted by *Bazzan & Grunitzki (2016)* shows that, in an en route trip building approach, the learning rate $\alpha$ does not play a major role, while the discount factor $\gamma$ usually needs to be high in discounted future rewards, as it is the case here. Thus a value of $\alpha = 0.5$ suits our needs. We remark however that we have also played with this parameter.

As for the discount factor $\gamma$, we have performed extensive tests and found that a value of $\gamma = 0.9$ performs best.

For the epsilon-greedy action selection, empirical analysis pointed to using a fixed value of $\epsilon = 0.05$. This guarantees that the agents will mostly take a greedy action (as they only have a 5% chance to make a non-greedy choice), and also take into account that the future rewards have a considerable amount of influence in the agent's current choice, since $\gamma$ has a high value.

For the bonus at the end of each trip, after tests, a value of $B = 1,000$ was used. Recall that this bonus aims at compensating the agent for selecting a jammed link, if it is close to its destination, rather than trying out detours via a link that, locally, seems less congested, but that will lead the agent to wander around, rather than directly go to its destination. We remark that trips take a rough average of 450 time steps thus this value of $B$ fits the magnitude of the rewards.

### Performance metric and results

While each agent perceives its own travel time, both after traversing each link, and after finishing its trip, we need an overall performance to assess the quality of the proposed

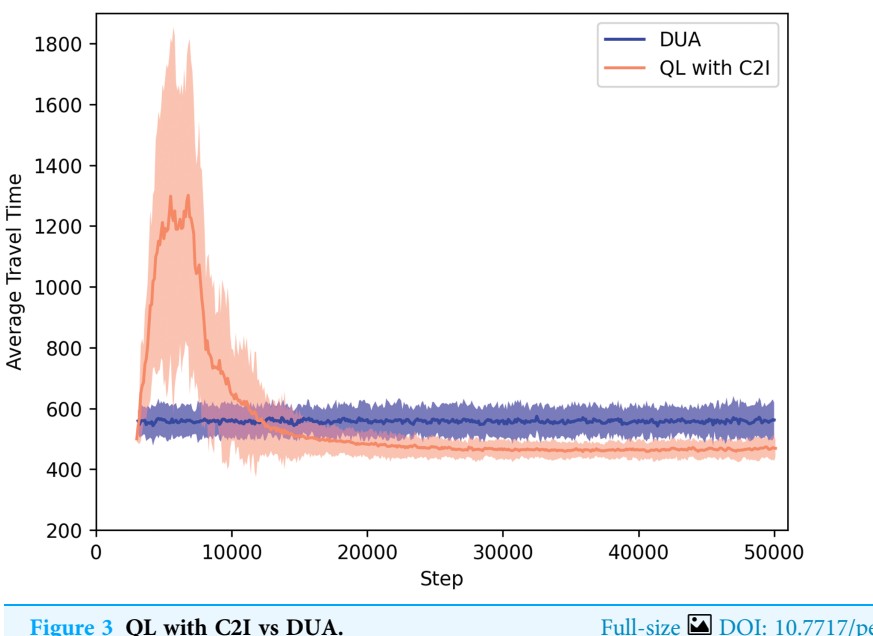

**Figure 3  QL with C2I vs DUA.**     

method. For this, we use a moving average (over 100 time steps) of the complete route travel time, for each agent that has finished its trip.

Given the probabilistic nature of the process, it is necessary to run repetitions of simulations. Thus, 30 runs were performed. Plots shown ahead thus depict the average and the standard deviations. In order to evaluate how the communication affects the learning process, some different policies and comparisons were performed, these different methods are described in the following sections.

### QL with C2I vs dynamic user assignment

For sake of contrasting with a classical approach, Fig. 3 shows a comparison between our QL with C2I approach and a method called Dynamic User Assignment (DUA), which is an iterative method implemented by the SUMO developers. We remark that DUA is a centralized, not based on RL approach.

Dynamic User Assignment works as follows: it performs iterative assignment of pre-computed, full routes to the given OD-pairs in order to find the UE[2]. In our tests, DUA was run for 100 iterations. Note that a DUA iteration corresponds to a trip, and a new iteration only starts when all trips have reached their respective destinations. The output of DUA is a route that is then followed by each vehicle, without en route changes. Since DUA also has a stochastic nature, our results correspond to 30 repetitions of DUA as well.

Figure 3 shows that, obviously, at the beginning, the performance of our approach reflects the fact that the agents are still exploring, whereas DUA has a better performance since a central authority determines which route each agent should take. This is possible since this central authority holds all the information, which is not the case in the MARL based approach, where each agent has to explore in order to gain information.

In our approach, after a certain time, the agents have learned a policy to map states to action and, by using it, they are able to reduce their travel times.

[2] For details on how the DUA method is made the reader may refer to https://sumo.dlr.de/docs/Demand/Dynamic_User_Assignment.html.

**Table 2 Travel time measured for DUA and QL with C2I at time step 50,000.**

| Method | Travel time at step 50 k |
|---|---|
| DUA | ≈560 |
| QL with C2I | ≈470 |

Before discussing the actual results, we remark that a SUMO time step corresponds roughly to one second. Our experiments were run for about 50,000 time steps. A learning episode comprehends hundreds of time steps, as the agent has to travel from its origin to its destination. In short, a learning episode is not the same as a simulation time step. Given that the agents re-start their trips immediately, different agents have different lengths for their respective learning episodes, thus the learning process is non-synchronous. Using our approach, on average, an episode takes roughly 500 time steps, thus agent reach the user equilibrium in about 100 episodes. For RL standards, this is a fast learning process, especially considering that we deal with a highly non-stationary environment, where agents get noisy signals. However, we also remark that, for practical purposes, the policy can be learned off-line, and, later, embedded in the vehicle.

To give a specific picture, Table 2 shows the actual travel times after time step 50,000. We remark that we could have measured roughly the same around step 30,000. It can be seen that our approach outperforms DUA shortly after time step 10,000. Also noteworthy is the fact that, at any time step, agents still explore with probability ε = 5% thus there is room for improvements if other forms of action selection are used.

### QL with C2I vs QL without communication

Our approach is also compared to standard Q-learning, thus without communication, which means that the agents learn their routes only by their own previous experiences, without any augmented knowledge regarding the traffic situation and the experiences of other agents.

In Fig. 4, we can divide the learning process in both cases shown in Fig. 4 in two distinct phases: the exploration phase, where the agents have yet no information about the network and explore it to find their destination "that is when the spikes in the learning curves can be seen"; and the exploitation phase, when agents know the best actions to take in order to experience the lowest travel time possible.

Both approaches converge to the same average travel times in the exploitation phase. However, the advantage of our approach comes in the exploration phase. As we see in Fig. 4, the exploration phase in the QL with C2I algorithm is reduced by a considerable amount when compared to the traditional QL algorithm, meaning that in our case the user equilibrium is reached earlier.

Table 3 compares the travel time measured in both cases at the time step 20,000, when our approach has already converged, but the standard Q-learning has not.

### Communication success rate

In the real world, it might be the case that some information gets lost due to failure in the communication devices. In order to test what happens when not all messages reach the

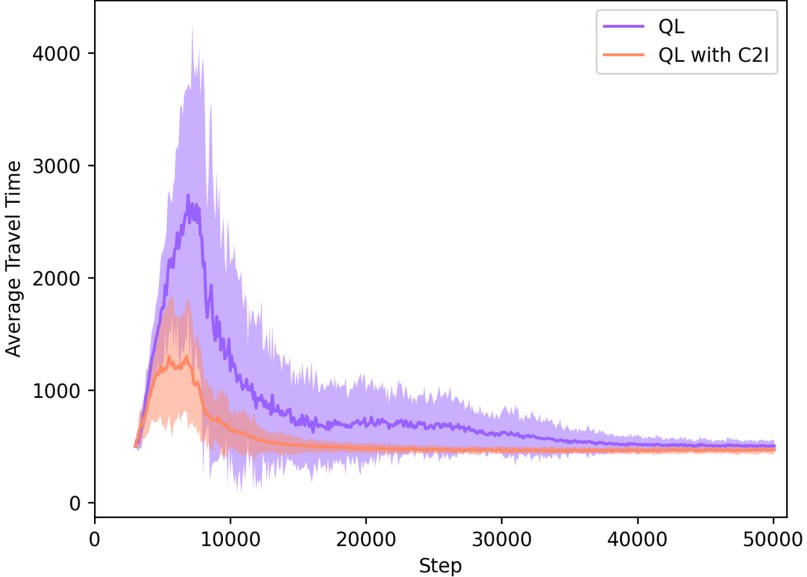

**Figure 4 QL with C2I vs QL without communication.**

**Table 3 Travel time measured for QL and QL with C2I at time step 20,000.**

| Method | Travel time at step 20k |
|---|---|
| QL | ≈676 |
| QL with C2I | ≈483 |

recipient, a success rate was implemented to test how the our approach performs if communication does not work as designed.

Specifically, every time an agent needs to communicate with the infrastructure, the message will reach the destination with a given success rate. This was implemented by means of a randomly generated value, which is then compared to the success rate to determine whether or not the simulator should ignore the message, thus being a metaphor for a non-delivered message. Such a scheme is applied to any kind of communication between the infrastructure and the agent, that is, regardless if it is from an agent to a CommDev, or vice-versa.

If a message is lost, then: (i) a CommDev does not get to update its data structure, and (ii) an agent does not get to update its Q-table. Other than that, the method behaves exactly as described by Algorithm 1.

Experiments were performed varying the target success rate. For clarity, we show the results in two plots.

Figure 5 compares the approach when the success rate is with 100% (thus the performance already discussed regarding the two previous figures), to one where the communication succeeds in only 75% of the times. In Fig. 6, we depict the cases for success rate of 25% and 50%.

For specific values, Table 4 lists the average travel times for all these cases at time step 20,000, since at that time the learning processes have nearly converged.

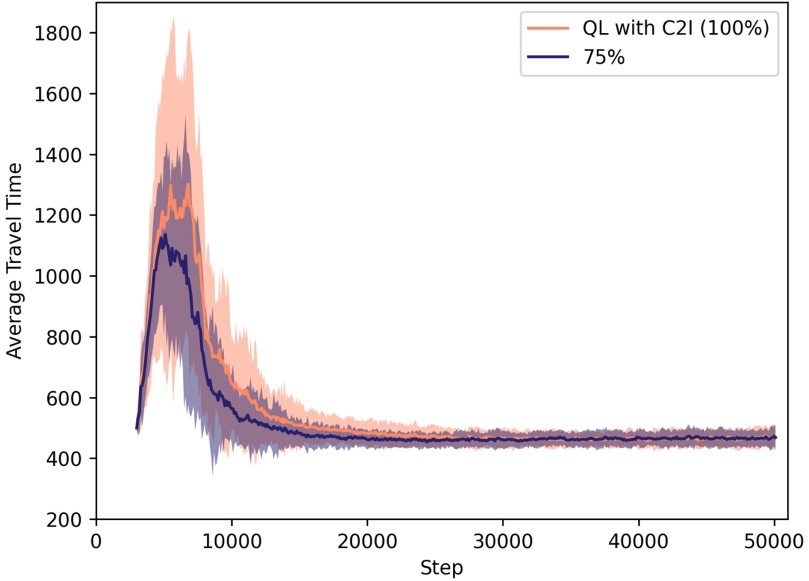

**Figure 5 QL with C2I: comparison between 75% and 100% success rate.**

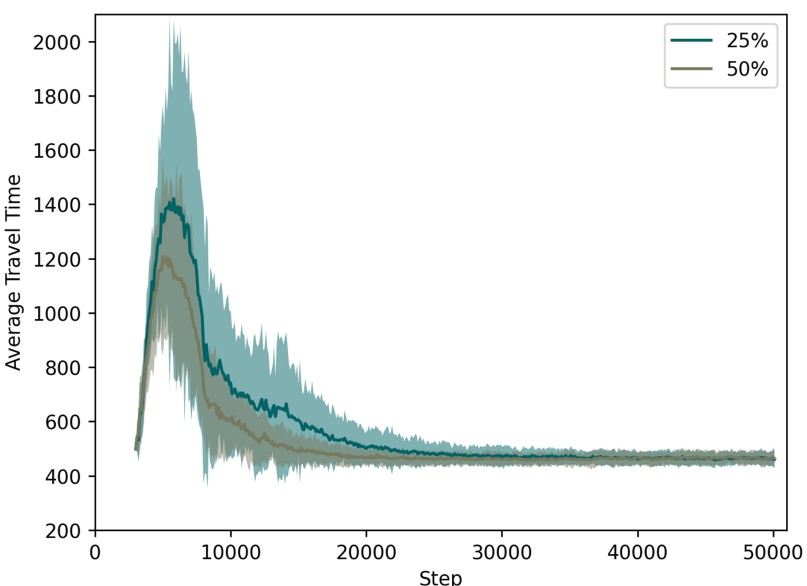

**Figure 6 QL with C2I: comparison between 25% and 50% success rate.**

It is remarkable that the system not only tolerates some loss of information, but also performs slightly better when the success rate is 75% or even 50%. If one compares this case to the one in which 100% of the messages reach their destinations, one sees that the learning process is accelerated if agents do not have the very same information that other agents also receive. This is no surprise, as pointed out in the literature on the disadvantages of giving the same information to everyone. What is novel here is the fact that we can show that this is also the case when information is shared only at local level, as

**Table 4 Travel time measured for each success rate at time step 20,000.**

| Success rate (%) | Travel time at step 20k |
|---|---|
| 25 | ≈501 |
| 50 | ≈467 |
| 75 | ≈461 |
| 100 | ≈483 |

well as when the communication is between vehicles and the infrastructure, not among all vehicles themselves.

As expected, when we look at the case with a low success rate of 25%, we observe several drawbacks since the communication rate is getting closer to no communication at all: (i) the average travel time increases, (ii) the learning process takes longer, and (iii) the standard deviation also increases (meaning that different trips may take very different travel times and, possibly, different routes).

### Different strategies for storing information at the infrastructure

Apart from investigating what happens when information is lost, we also change the way CommDevs compute and share the reward information to the driver agents. Here the main motivation was to test what happens when the infrastructure is constrained by a simpler type of hardware, namely one that can store much less information (recall that the original approach is based on a queue-like data structure).

To this aim, we conducted experiments in which the goal was to test which type of information is best for the infrastructure to hold and pass on to the agents. We have devised three ways to do this: (i) the infrastructure only holds and informs the highest travel time (hence the most negative reward) value to the agents; (ii) the infrastructure informs the lowest reward (hence the least negative) to the agents; (iii) the infrastructure holds only the latest (most recent) travel time value received. Note that, in all these cases, the infrastructure only needs to store a single value, as opposed to the case in which the infrastructure stores a queue of values in order to compute a moving average.

Figure 7 shows a comparison between the different policies. For clarity, we omit the deviations but note that they are in the same order as the previous ones.

The best case seems to be associated with the use of the most recent travel time information, as seen both in Fig. 7 and Table 5. Communicating the lowest travel time might look good at first sight. But it has as drawback that this leads all agents to a act greedily and thus using the option with least travel time. This ends up not being efficient as seen in Fig. 7. Conversely, communicating the highest travel time is motivated by the fact that the infrastructure might want to distribute the agents among the options available, thus communicating a high travel time leads to not all agents considering it: since some would have experienced a better option before and hence have this knowledge in their Q-tables, they will not use the information received. This proves to be the second best strategy, only behind the aforementioned strategy on communicating the latest information. The reason for the good performance of the latter is the fact that the latest

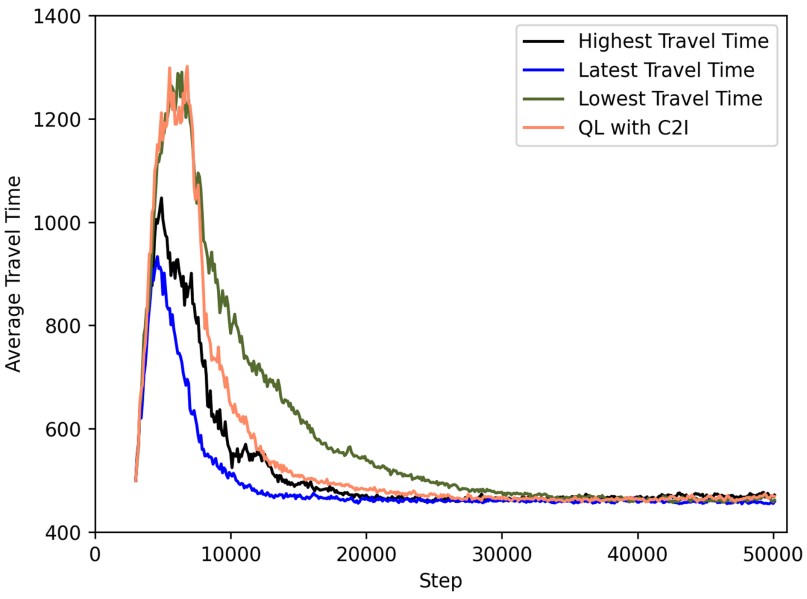

**Figure 7 QL with C2I with different strategies.**

**Table 5 Travel time measured for each strategy at time step 20,000.**

| Strategy | Travel time at step 20 k |
| --- | --- |
| Highest travel time | ≈472 |
| Latest travel time | ≈467 |
| Lowest travel time | ≈538 |
| QL with C2I | ≈483 |

information is diverse enough (i.e., varies from recipient agent to agent) so that it also guarantees a certain level of diversity in the action selection, thus contributing to a more even distribution of routes.

## CONCLUSIONS AND FUTURE WORK

A wise route choice is turning more and more important when the demand is increasing and road networks are not being expanded in the same proportion. MARL is an attractive method for letting agents autonomously learn how to construct routes while they are traveling from A to B.

This article presented a method that combines MARL with C2I communication. Vehicles interact with the infrastructure every time they reach an intersection. While they communicate the travel times they have experienced in nearby links, they also receive the expect travel times regarding their next possible link choices. We have extended a previous approach by relaxing the assumption that all messages are sent and received, that is, there is no loss of messages. To the best of our knowledge, this is a novel investigation to scenarios dealing with learning based route choice, where the there is a sharing of local information via C2I.

This work thus has the following contributions: we employ MARL to the task of learning; we do this using a non trivial scenario with more than one origin-destination pair; we depart from the assumption that driver agents already know a set of (pre-computed) routes to select among; we use a microscopic, agent-based approach; we connect MARL with new communication technologies, in order to investigate whether the learning process can be accelerated. Also, we have employed our method to test some real-world situations that may arise, namely communication loses and the need to use simpler hardware devices to store information by the infrastructure.

Our results show that, before deploying C2I communication in the real-world, one has to take into account the various effects of sharing information, even at local level. We were able to show that one has to strive to communicate information that is diverse enough, in order to avoid sub-optimal route choices, that is, those that are made by drivers having similar information. As these drivers tend to act greedly, a wise strategy on sharing information is key.

Specifically, our results point out to our approach being tolerant to information loses; further, there was even a slight improvement in the overall performance (i.e., learning speed) since less information also mean that not all agents will act the same way. As for the different strategies regarding storage of information in the infrastructure, we could show that communicating only the latest known travel time is able to speed up the learning process.

We remark that in all cases we have tested, MARL was able to reach the user equilibrium. The major difference is the speed of such process.

For future work, one possible investigation is the addition of a biased information provided by the infrastructure in order to reach a different outcome, namely, to reach the system optimum (socially efficient distribution of routes to vehicles), rather than converging to the user equilibrium. We also plan to change the demand during simulation time, to check how the learners deal with such changes. Preliminary work on using Q-learning in such dynamic environments point out that it is able to handle different situations. However, it remains to be investigated whether this is also the case for changes in flow rates. Moreover, we would like to study whether the proposed combination of Q-learning with C2I is able to speed up the learning processes as much as it was the case in the present work.

### Funding

This work was supported by CNPq under grant no. 307215/2017-2 (Ana Bazzan), by CAPES (Coordenação de Aperfeiçoamento de Pessoal de Nível Superior—Brazil, Finance Code 001), and by a FAPERGS grant (Guilherme D. dos Santos). There was no additional external funding received for this study. The funders had no role in study design, data collection and analysis, decision to publish, or preparation of the manuscript.

## Grant Disclosures

The following grant information was disclosed by the authors:
CNPq: 307215/2017-2.
CAPES (Coordenação de Aperfeiçoamento de Pessoal de Nível Superior—Brazil): 001.
FAPERGS.

## Competing Interests

The authors declare that they have no competing interests.

## Author Contributions

- Guilherme Dytz dos Santos conceived and designed the experiments, performed the experiments, analyzed the data, performed the computation work, prepared figures and/or tables, authored or reviewed drafts of the paper, and approved the final draft.
- Ana L.C. Bazzan conceived and designed the experiments, analyzed the data, prepared figures and/or tables, authored or reviewed drafts of the paper, and approved the final draft.

## Data Availability

Code is available at GitHub: https://github.com/guidytz/SUMO-QL.

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
