# Peer review of "Sharing diverse information gets driver agents to learn faster: an application in en route trip building"

_PeerJ Computer Science, doi:10.7717/peerj-cs.428_

## Round 0.1 · original submission · Major Revisions

Both reviewers found the paper interesting and with potential, but they suggest several improvements that I also consider relevant.

Reviewer 1 ·

Basic reporting

The paper reports on very relevant and contemporary topics to both computer science and ITS (intelligent transportation systems). It is, therefore, a very timely and significant article. It studies the effect of providing driver agents with current congestion information through vehicle-2-infrastructure strategies, and adapt Q-Learning to support their approach of a decentralised traffic assignment method.

The article is written and organised well, quite clear and easy to follow. Some typos are found, nonetheless are very easily addressed with careful proofreading (e.g. "We could shown that...").

Some concepts could be better explained and interpreted, though. It is not very clear why authors do not consider their proposed method as a "traffic assignment" method, as discussed in line 49 and on. After all, the proposed method is an iterative offline method that seeks to yield user equilibrium. In which sense is it not a traffic assignment problem then?

Experimental design

The proposed method is described well, and the experimental setup is appropriate to support the evaluation of the authors' approach.

Some issues on the definition of the network might be further explained, though:

1) How is the impedance of intersections modelled? Are intersections traffic-light controlled or are they priority junctions?

2) How dense is the graph representing the communication network? Are all CommDev connected to each other (i.e. |E| ~ V^2)? What would be the impacts of different assumptions for this graph on the results?

Also, please provide some more details on how this work extends the one previously published by the same authors, quite recently: (Santos & Bazzan, 2020). Much material in this paper seems to come as-is from the previous publication, especially concerning the proposed method.

Validity of the findings

Results seem to be good and very promising. Indeed, gains in the learning speed of RL drivers are significant. However, some findings seem very expected and intuitive; also the fact that providing all drivers with information won't benefit the system performance and considering a lower rate of informed drivers produces better outcomes, even in real-time setups (for rather localised information, in space and in time). Are there other findings that may stimulate the potential use of the proposed approach?

Additional comments

The subjects addressed in this paper are indeed timely and significant. As expected, information can improve and speed up the learning process of RL drivers resorting to two-way communication technologies as preconised in ITS directives. It is not very clear, however, how is this method different from other user-equilibrium offline iterative methods, regarding the concept of traffic assignment as a tool for planning, rather than for operational purposes.

Reviewer 2 ·

Basic reporting

This paper is very well written. ALL the contexts are clearly expressed. The references are up-to-date.

Experimental design

The experimental should be strengthened by providing comparison between their method and some well-known methods to show the novelty.

Validity of the findings

The novelty of the findings should be strengthened.

Additional comments

This paper devises a Q-leaning scheme for en-route trip building problem. After reading this paper, I think it is well written, but needs further improvement to clarify the novelty of findings.

1. The model of TAP should be introduced.

2. The comparison between their method and some well-known RL schemes should be provided to show the novelty.

So, I recommend this paper to be accepted as major.

---

## Round 0.2 · accepted · Accept

One of the reviewers has found the new version of the paper suitable. The other reviewer is having some problems to revise this second version of the manuscript. I have carefully checked your response to the issues raised by the reviewer and I consider them appropriate.

Reviewer 2 ·

Basic reporting

No comment

Experimental design

no comment

Validity of the findings

no comment

Additional comments

I am satisfied with the revised version, since all my concerns have been addressed. So, I recommend this paper to be accepted for publication.